# PI3KC2α-dependent and VPS34-independent generation of PI3P controls primary cilium-mediated autophagy in response to shear stress

Asma Boukhalfa[1,4], Anna Chiara Nascimbeni[1,4], Damien Ramel [2], Nicolas Dupont[1], Emilio Hirsch [3], Stephanie Gayral[2], Muriel Laffargue[2]*, Patrice Codogno[1]* & Etienne Morel [1]*

Cells subjected to stress situations mobilize specific membranes and proteins to initiate autophagy. Phosphatidylinositol-3-phosphate (PI3P), a crucial lipid in membrane dynamics, is known to be essential in this context. In addition to nutriments deprivation, autophagy is also triggered by fluid-flow induced shear stress in epithelial cells, and this specific autophagic response depends on primary cilium (PC) signaling and leads to cell size regulation. Here we report that PI3KC2α, required for ciliogenesis and PC functions, promotes the synthesis of a local pool of PI3P upon shear stress. We show that PI3KC2α depletion in cells subjected to shear stress abolishes ciliogenesis as well as the autophagy and related cell size regulation. We finally show that PI3KC2α and VPS34, the two main enzymes responsible for PI3P synthesis, have different roles during autophagy, depending on the type of cellular stress: while VPS34 is clearly required for starvation-induced autophagy, PI3KC2α participates only in shear stress-dependent autophagy.

[1] Institut Necker-Enfants Malades (INEM), INSERM U1151 CNRS UMR 8253, Université Paris Descartes-Sorbonne Paris Cité, Paris, France. [2] Institute of Metabolic and Cardiovascular Diseases, INSERM UMR 1048, Paul Sabatier University, Toulouse, France. [3] Molecular Biotechnology Center, Department of Molecular Biotechnology and Health Sciences, University of Torino, Turin, Italy. [4]These authors contributed equally: Asma Boukhalfa, Anna Chiara Nascimbeni. *email: muriel.laffargue@inserm.fr; patrice.codogno@inserm.fr; etienne.morel@inserm.fr

In the phosphoinositides family, the phosphatidylinositol-3-phosphate (PI3P) is a central lipidic player in membrane dynamics and trafficking regulation in eukaryotic cells[1,2]. PI3P is mainly synthesized by the class 3 PI3 Kinase (PI3KC3, also known as VPS34) and is associated with the autophagic machinery[3–6] and the endosomal functions[7,8], both systems being connected with the lysosome, the acidic degradative organelle in eukaryotes. While the endosomal system is mostly responsible for sorting, recycling, and degradation of plasma membrane components and external material[9,10], autophagic membranes and their associated machinery are mobilized during stress situations to ensure the degradation of intracellular components by triggering the formation of autophagosomes, which capture cytoplasmic material and subsequently lead it to degradation upon fusion with lysosomes[11–13]. The formation of these double membrane organelle is initiated by two ATG (autophagy-related) containing complexes: the ULK signaling complex and the PIK3C3 complex I. The core of the latter is composed by Beclin1, ATG14L, VPS15, and the lipid kinase VPS34. When associated with the above-mentioned partners, VPS34 produces PI3P at autophagosomal membrane source sites allowing the recruitment of autophagy-associated PI3P-binding proteins such as DFCP1 and WIPI-2, and thus directly participates in the activation of the downstream ATG machinery, including the lipidated LC3 (LC3.II), which leads to the nucleation of the autophagosomal membrane[14]. Among other cellular structures that participate to external stress sensing, the primary cilium (PC), an essential microtubule-based organelle located at the apical side of most epithelial cells[15], triggers a signaling cascade upon sensing extracellular chemical and mechanical stimuli[16], including shear stress. Interestingly, we have recently shown that autophagic machinery is also stimulated in response to fluid-flow-induced shear stress in kidney epithelial cells, a situation that requires PC signaling and leads to cell size regulation[17,18]. Finally, the PI3KC2α lipid kinase, considered as an alternative source of PI3P, has been shown to participate in the regulation of PC biogenesis and PC associated signaling via specific PI3P synthesis in embryonic fibroblasts and renal epithelial cells[19,20]. Altogether, these observations combining PC, PI3P and PI3KC2α, highlight a putative role for this kinase in the intriguing connection between autophagy and PC function in response to mechanical stimulation.

Here we demonstrate that the lipid enzyme PI3KC2α, whose expression is increased during shear stress, is responsible for the synthesis of a specific pool of PI3P at the vicinity of the PC in response to fluid flow in kidney epithelial cells. We show that the knockdown of PI3KC2α not only abrogates shear-stress-induced PI3P production, but also abolishes autophagy as well as cell volume adaptation to shear stress. Similar data were observed in PI3KC2α$^{+/-}$ mice epithelial kidney cells in vivo. The specific need for PI3P production in response to shear stress was further demonstrated in cells lacking a functional PC in which the autophagy program associated with shear stress was restored upon artificial delivery of exogenous PI3P. We finally show that while PI3KC3/VPS34 is, as expected, required for starvation-induced autophagy, it is not involved in shear-stress-associated autophagy. In sharp contrast, PI3KC2α is specifically required for the autophagic program associated with mechanical stimulation relayed by the PC, but not essential for nutritional stress response, suggesting that a selective pool of PI3P is generated during shear stress.

## Results

### Shear stress induces a PI3KC2α-associated PI3P pool at PC.
We previously reported that a constant laminar fluid flow on mouse proximal tubule kidney epithelial cells induces PC-dependent autophagy activation and cell size regulation[17]. Similar results were obtained using human HK2 cells (proximal tubule kidney cells) (Supplementary Fig. 1 and[18]). We indeed show that, compared to a static cell culture situation, a 4 days-long 1 dyn/cm$^2$ laminar fluid flow induces an autophagic response, as shown by an increase in LC3-II (Supplementary Fig. 1a) and in the number of LC3-positive structures in polarized cells during 4 days (Supplementary Fig. 1b, c). Moreover, shear stress induces a decrease in cell size (Supplementary Fig. 1b, d) as previously shown in mice kidney epithelial cells and MDCK cells[17,21]. We observed as well that shear stress increases PI3KC2α stability at the protein level (Fig. 1a, b) and mRNA level (Fig. 1c). It also leads to a direct and local mobilization of PI3KC2α at the PC (Fig. 1d, e), indicating that PI3KC2α and the autophagic machinery (Supplementary Fig. 1) are both reactive to shear stress in ciliated cells. As the main known enzymatic function of PI3KC2α is to synthesize PI3P[22], we tested whether the shear stress response could also mobilize a dedicated pool of PI3P membranes at the PC vicinity. We thus monitored and quantified the local changes in PI3P-positive structures at the PC basal body area (see scheme on Fig. 1f) in fixed cells using indirect fluorescent recombinant FYVE dye to avoid overexpression of PI3P-binding domains[23] and ARL13B as a specific marker for PC axoneme. In accordance with PI3KC2α mobilization upon fluid-flow stimulation, we show that shear stress also induces the appearance of a local pool of PI3P at the base of the PC (Fig. 1g, h).

### PI3KC2α knockdown affects shear stress-associated PI3P.
We analyzed next the effect of PI3KC2α knockdown (PI3KC2α siRNA, Supplementary Fig. 2a) on polarized epithelial cells submitted to shear stimulation. While the siRNA-mediated knockdown (Supplementary Fig. 3a) of VPS34, the other major enzyme responsible for PI3P synthesis, has no effect on PC (Supplementary Fig. 3b, c) we show that PI3KC2α depletion abrogates ciliogenesis (Supplementary Fig. 2b–d), as previously reported[19]. Moreover, we show that the AMPK/LKB1 signaling cascade[21,24], which is associated with fluid flow sensing by the PC[17,18], is abolished in PI3KC2α KD cells, since p-AMPK (Supplementary Fig. 4a, b) and LKB1 (Supplementary Fig. 4c, d) recruitment to PC are compromised in the absence of PI3KC2α. To analyze precisely the dynamics of the PI3P present in the basal body area (see Fig. 1g), we systematically quantified the PI3P-positive structures in a 150 μm$^2$ circular area close to the nucleus (corresponding to the PC basal body location, Fig. 2a, b) in control cells and in PI3KC2α KD cells, in which PC presence was inhibited (Supplementary Fig. 2). Interestingly, while absence of PI3KC2α has no consequences on the number of PI3P structures in the PC area in static culture situation, the pool of PI3P-positive membranes drops by approximately 50% in cells lacking PI3KC2α during shear-stress treatment (Fig. 2a, c).

Importantly, the amount of autophagy-associated WIPI2 PI3P-binding protein[25] was increased upon shear stress but reduced in PI3KC2α KD cells (Fig. 2d), as well as WIPI2 mRNA levels (Fig. 2e), arguing for a specific autophagic response mediated by a PI3KC2α synthesized pool of PI3P in shear-stress treatment. Importantly, a pool of WIPI2 protein is associated with the PI3P observed at the base of PC in response to shear stress (Fig. 2f). Moreover, the small GTPase Rab11a, a key partner of WIPI2 in autophagy[26,27] and PI3P-positive membranes in relationship with PC and endocytosis turnover regulation[19,28], was increased by shear-stress stimulation (Supplementary Fig. 5a, b). In addition, we observed that the GTP bound (i.e. activated) form of Rab11a was partially addressed to the cilium axoneme in the same situation ($^{GTP}$Rab11, Supplementary Fig. 5c). Interestingly, the knockdown of PI3KC2α inhibits this Rab11a behavior in response

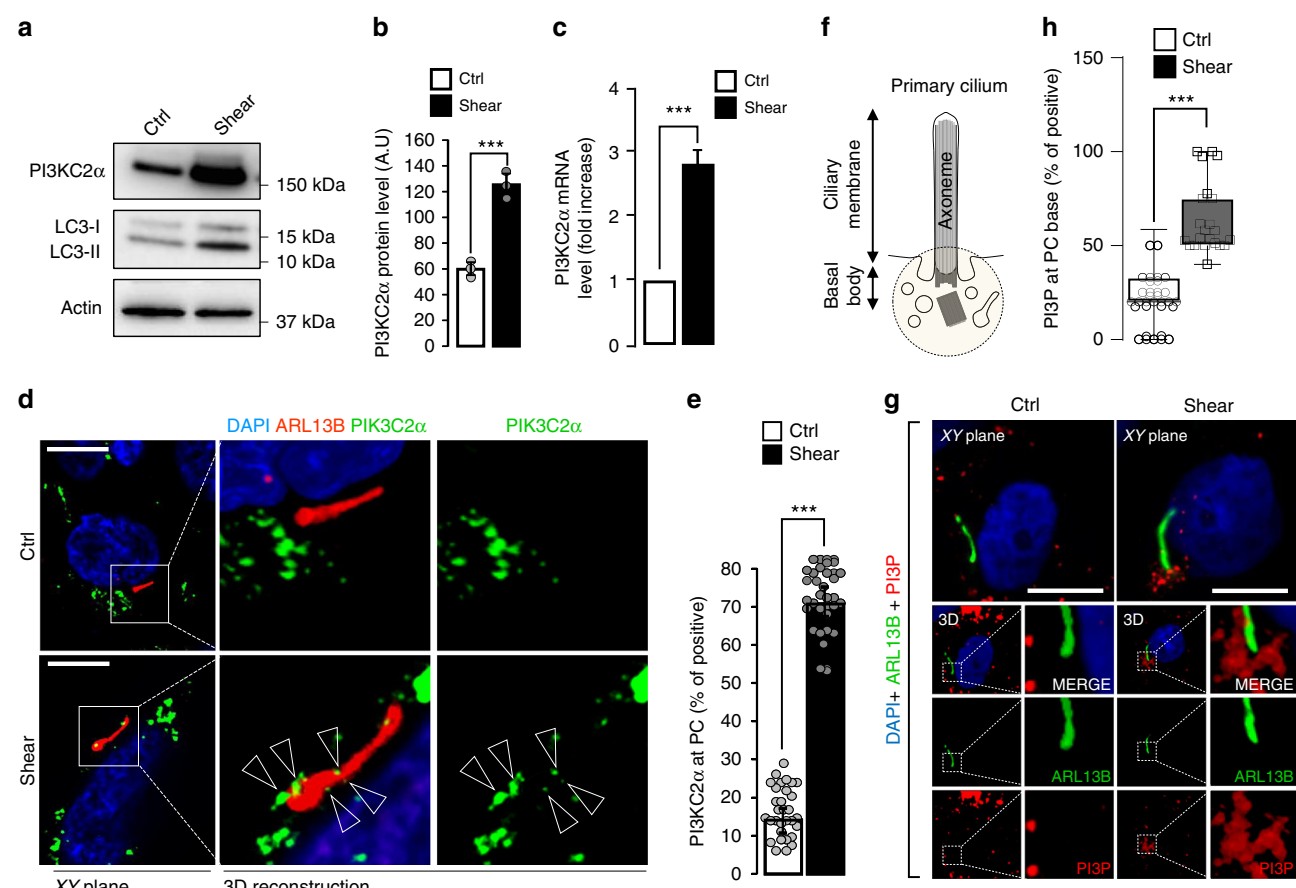

**Fig. 1 Shear stress induces PI3KC2α expression and a local pool of PI3P at the primary cilium. a, b** Western blot analysis of LC3-I, LC3-II and PI3KC2α and quantification of PI3KC2α protein (**b**) and mRNA levels (**c**) in lysates of polarized HK2 cells subjected to shear stress for 96 h, compared to static cultured cells (ctrl, 96 h). Bar graphs denote average protein levels normalized to actin (**b**) (mean ± SEM, from three independent experiments) and fold increase of average mRNA levels (**c**) relative to ctrl and normalized to actin mRNA (mean ± SEM, from three independent experiments). ***$p < 0.001$ in two-tailed Student's $t$ test. **d**, **e** Representative 3D confocal acquisitions (**d**) and quantification (**e**) of polarized HK2 cells subjected to shear stress (96 h), compared to static cultured cells (ctrl, 96 h), immunostained for PI3KC2α, ARL13B (primary cilium (PC) marker) and DAPI, showing increased PI3KC2α expression at the PC upon shear stress ($N = 30$ cells, from three independent experiments). Scale bar, 10 μm. **f** Schematic representation of PC structure and zone (150 μm² circular area) of PI3P-positive structures quantification. **g, h** Representative 3D confocal acquisitions and related PI3P quantification of polarized HK2 cells subjected to shear stress (96 h), compared to static cultured HK2 cells (ctrl, 96 h), immunostained for the primary cilia (PC) axoneme marker ARL13B and for PI3P (using indirect FYVE-GST recombinant peptide), showing PI3P-positive structures at the base of PC area (mean ± SEM, $N = 100$ cells, from three independent experiments). ***$p < 0.001$ in two-tailed Student's $t$ test. Scale bar, 10 μm.

to fluid flow, as revealed by biochemical analysis (Supplementary Fig. 5a, b) and immunofluorescence (Supplementary Fig. 5c). Finally, in conditions of PI3KC2α depletion, we analyzed the behavior of ATG16L1, a crucial regulator of early autophagic processes, notably as a key partner of WIPI2 and PI3P in autophagy regulation[29,30], and known to be recruited to the basal body in PC associated autophagy[31] during shear stress[18]. Importantly we showed that, as observed for Rab11a and WIPI2, ATG16L1 was mobilized by shear stress but not in siPI3KC2α transfected cells (Supplementary Fig. 6a, b). The local recruitment of ATG16L1 at the base of PC observed in control cells was no longer detectable in PIK3C2a knockdown cells (Supplementary Fig. 6c, d), highlighting the crucial role of the autophagy-related PI3P-associated machinery in response to shear stress.

Altogether, these results suggest that shear stress-induced PI3P mobilization and autophagy both depend on PI3KC2α lipid kinase.

**PI3KC2α, but not VPS34, regulates shear stress-associated autophagy and cell size regulation.** *PI3KC2α and VPS34 have*

*different roles in autophagy*: As shown in supplementary Fig. 1, shear-stress treatment of polarized epithelial cells leads to autophagy induction and cell size decrease. PI3P measurements by fluorescence (Fig. 2) suggest that PI3KC2α is responsible for shear stress-induced PI3P synthesis. However, VPS34 (also known as PI3KC3) has been described as the canonical lipid kinase regulating PI3P-associated autophagy[14,32]. To decipher the precise function(s) of PI3KC2α in our experimental system, we thus compared autophagy elicited by shear stress in cells KD for PI3KC2α or PI3KC3/VPS34. Interestingly, we observe that LC3 lipidation induced by shear stress is blocked by PI3KC2α KD but not by VPS34 KD (Supplementary Fig. 7a, b), suggesting that shear-stress-associated autophagy is solely dependent on PI3KC2α, and not VPS34. Next we compared the number of LC3-positive endomembranes in cells KD for VPS34 or PI3KC2α and subjected to either 24 h shear stress or 24 h nutritional stress (starvation), both situations inducing autophagy. Surprisingly, and in accordance with western blot data, we show that the presence of PI3KC2α was necessary for membrane LC3 mobilization upon shear stress, but not upon starvation (Fig. 3a, b, siPI3KC2α conditions). In contrast, while VPS34 is required for starvation-

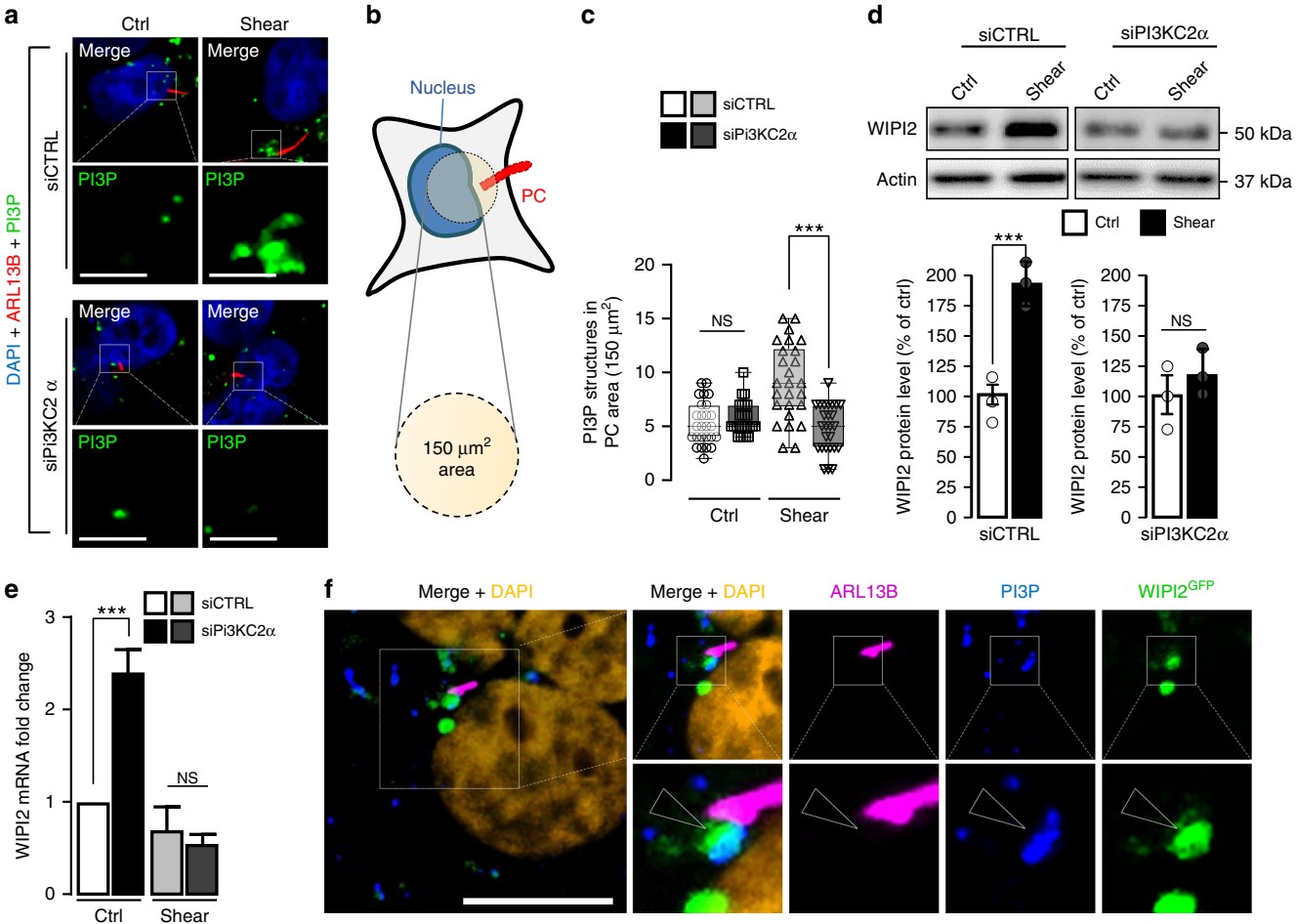

**Fig. 2 Shear-stress-induced PI3P synthesis at the PC depends on PI3KC2α. a** Representative confocal images upon shear-stress conditions of PI3KC2α knocked down HK2 cells (siPI3KC2α), compared to control cells (siCTRL), immunostained for ARL13B, PI3P (using FYVE-GST indirect recombinant peptide) and DAPI ($N = 80$ cells, from five independent experiments). Scale bar, 10 μm. **b** Schematic drawing showing the 150 μm² circular area centred at the cell nucleus used as identifier of the PC basal body area. **c** Quantification of the PI3P-positive structures at the PC area, as defined in (**b**) and shown in (**a**), in siCTRL or siPI3KC2α HK2 cells, upon static (ctrl) and shear-stress (96 h) conditions (mean ± SEM, $N = 80$ cells, from five independent experiments). NS: not significant, ***$p < 0.001$ in two-tailed Student's $t$ test. **d** Western blot analysis and quantification of WIPI2 protein levels in lysates of polarized siCTRL or siPI3KC2α HK2 cells, upon static (ctrl) and shear-stress (96 h) conditions. Bar graph denotes average protein levels normalized to actin (mean ± SEM, from three independent experiments). NS: not significant, ***$p < 0.001$ in two-tailed Student's $t$ test. **e** WIPI2 mRNA levels quantifications from total lysates of polarized siCTRL or siPI3KC2α HK2 cells, upon static (ctrl) and shear-stress (96 h) conditions. Bar graphs denote fold change of average mRNA levels relative to ctrl and normalized to actin mRNA (mean ± SEM, from three independent experiments). ***$p < 0.001$ in two-tailed Student's $t$ test. **f** Representative confocal acquisition of WIPI2^GFP transfected HK2 cells upon shear-stress conditions immunostained for ARL13B, PI3P (using FYVE-GST indirect recombinant peptide) and DAPI showing PI3P and WIPI2^GFP colocalization at the base of PC (arrowhead). Scale bar, 10 μm.

induced LC3 mobilization, its depletion had no consequence on the shear stress-induced LC3 membrane mobilization (Fig. 3a, b, siVPS34 conditions). Similar results were obtained when we quantified the total number of PI3P-positive cellular structures (i.e. early endosomal and autophagosomal membranes), in the very same conditions: while VPS34 is required for PI3P synthesis at steady state and during starvation-induced autophagy, absence of PI3KC2α is only deleterious for shear-stress-induced autophagy (Supplementary Fig. 7c, d), confirming the above mentioned LC3 data and illustrating the importance of PI3KC2α in shear stress-associated autophagic machinery mobilization.

Finally, we show that starvation-induced autophagy has no effect on the cell size decrease observed upon fluid-flow treatment and, more interestingly, that PI3KC2α KD completely abolishes this cell size adaptation while VPS34 KD has no effect on it (Fig. 3c, d). To confirm the non-involvement of the PI3KC3 complex in shear-stress-associated autophagy, we investigated Beclin1 behavior in response to mechanical stress, since it is a

master regulator of VPS34 activity regulation in autophagy-associated PI3P synthesis[33]. Interestingly, while PI3KC2α protein stability is associated with shear-stress response (Fig. 1a, b), Beclin1 turnover was not affected either by shear stress or by PI3KC2α knockdown (Supplementary Fig. 8a). In cells lacking Beclin1 (BECN1 siRNA, Supplementary Fig. 8b), LC3 lipidation induced by shear stress is comparable to the control cells (Supplementary Fig. 8c), illustrating that the absence of Beclin1 protein has no deleterious effect on shear stress-associated autophagy. This was confirmed, in cells subjected to shear stress, by the absence of a Beclin1 knockdown effect on quantifications of LC3-positive structures (Supplementary Fig. 8d, e) and on cell volume regulation (Supplementary Fig. 8f, g). Similar results were obtained when we questioned the involvement of FIP200, a key protein in starvation-induced autophagy, which regulates VSP34/Beclin1 activity with its partners ULK1/2 and ATG101[34,35]. We show that FIP200 amount was not affected by shear stress (Supplementary Fig. 9a, siCTRL conditions), unlike PI3KC2α and

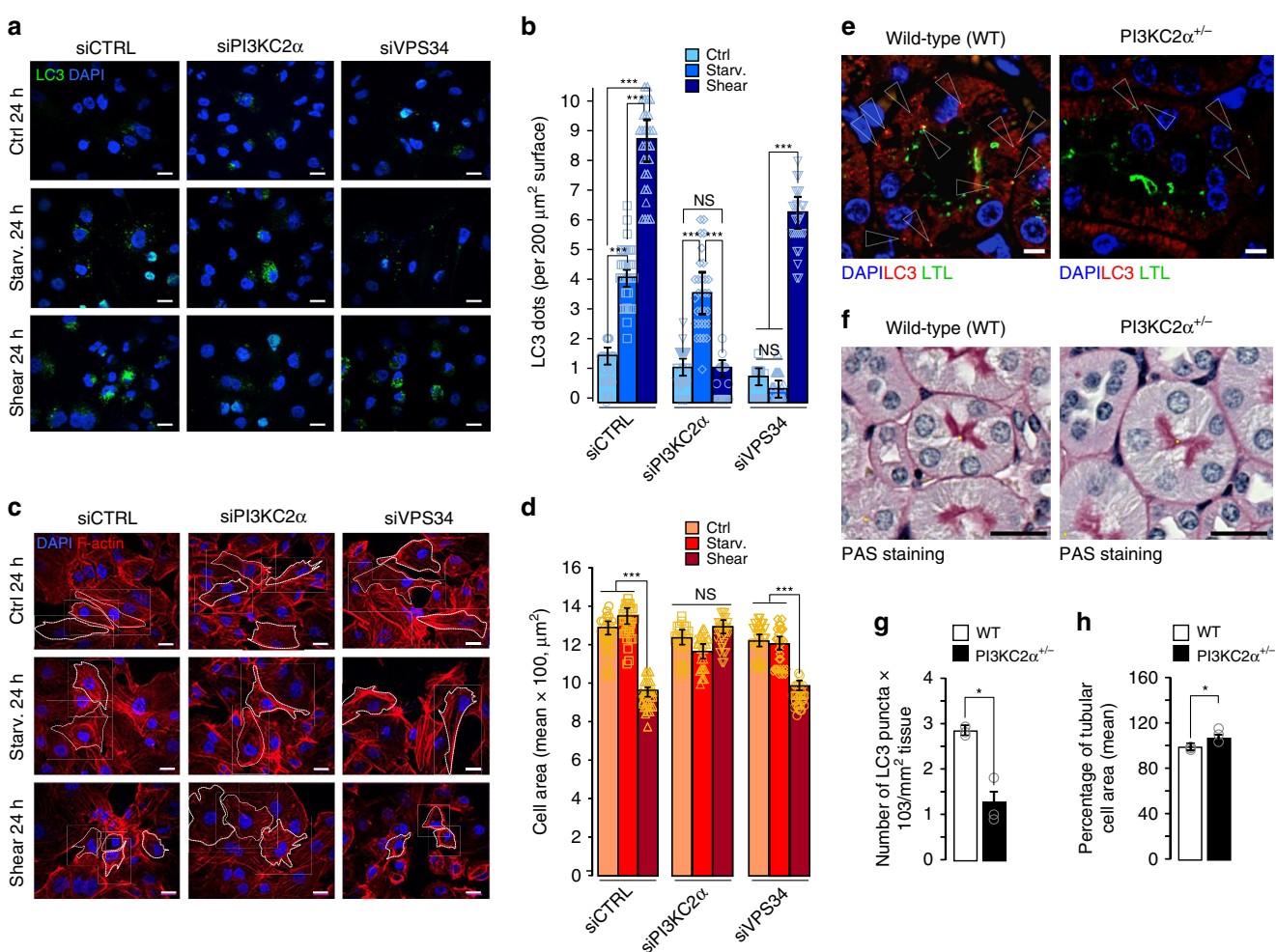

**Fig. 3 Shear-stress-induced autophagy and cell size regulation depend on PI3KC2α, but not on PI3KC3/VPS34. a**, **c** Representative confocal images of polarized HK2 cells (siCTRL), compared to PI3KC2α knocked down cells (siPI3KC2α) and PI3KC3/VPS34 knocked down cells (siVPS34), upon static (ctrl, 24 h), shear stress (24 h) and starvation conditions (24 h, see Methods section for more details), immunostained for LC3, F-actin and DAPI. Scale bar, 10 μm. **b**, **d** Quantifications of (**a**) and (**c**). Bar graphs denote average number of LC3 puncta and average cell areas (mean ± SEM, N = 100 cells, from three independent experiments). NS: not significant, ***p < 0.001 in two-tailed Student's t test. **e**, **g** Cortex zone of kidneys from wild type (WT) and PI3KC2$^{+/-}$ mice, immunostained for LC3, LTL (*Lotus Tetragonolobus* Lectin) and DAPI, and quantification of LC3 puncta (mean ± SEM, from three independent experiments). Scale bar, 10 μm. *p < 0.05 in two-tailed Student's t test. **f**, **h** PAS (Periodic Acid Schiff) staining of the cortex zone of kidneys from wild type and PI3KC2$^{+/-}$ mice and quantification of tubular cell area (mean ± SEM, from 3 independent experiments). Scale bar, 50 μm. *p < 0.05 in two-tailed Student's t test.

WIPI2. Moreover, the siRNA-mediated knockdown of FIP200 has no effect either on LC3 lipidation (Supplementary Fig. 9a, b) or LC3 dots numbers (Supplementary Fig. 9c, d) and cell size regulation (Supplementary Fig. 9e, f) during shear stress. These observations confirm that the classical physical and functional partners of VPS34 activity during starvation-induced autophagy are not required in shear-stress-associated autophagy.

Finally, our in vitro observations about PI3KC2α implication in fluid-flow-associated autophagy were confirmed in vivo by comparing wild type (WT) and PI3KC2α$^{+/-}$ mice in the cortex zone of kidneys. We observe a decrease in the LC3-positive structures number in the PI3KC2α$^{+/-}$ mice (Fig. 3e–g) as well as an increase in the cell surface compared to wild type (Fig. 3f–h). Altogether, these results show that PI3KC2α is required for autophagy induction and cell size regulation in cells subjected to shear stress.

**Exogenous PI3P delivery rescues autophagy in absence of PC.** Our results indicate that fluid flow mediates PI3KC2α

mobilization to induce PI3P synthesis at the vicinity of PC basal body (Fig. 1). We thus wondered whether artificial delivery of PI3P could counteract the absence of the PC (see experimental set-up in Fig. 4a), considered as the central signaling structure that triggers the response to shear stress[16]. To do this, we knocked down the PC associated IFT88 protein[36] (Fig. 4b) and we showed, as expected, that IFT88 siRNA transfected cells displayed altered ciliogenesis (Supplementary Fig. 10a, b), similarly to what we can observe for other IFT protein, such as IFT20 (Supplementary Fig. 10c–e). IFT88 knockdown cells failed to induce proper autophagic response to shear stress (Supplementary Fig. 10f, g) and interestingly, absence of IFT88 also alters PI3KC2α stability (Supplementary Fig. 10h, i). Finally, and making sense with alteration of PI3KC2α stability, we show that IFT88 knockdown cells failed to induce local and shear stress associated PI3P synthesis as we observed in control cells (Supplementary Fig. 10j, k). As expected IFT88 knockdown leads to the abolition of cellular adaptation to fluid-flow-induced shear stress, demonstrated by the lack of LC3-positive structures increase (Fig. 4c, d, siCTRL and siIFT88 + lipid carrier

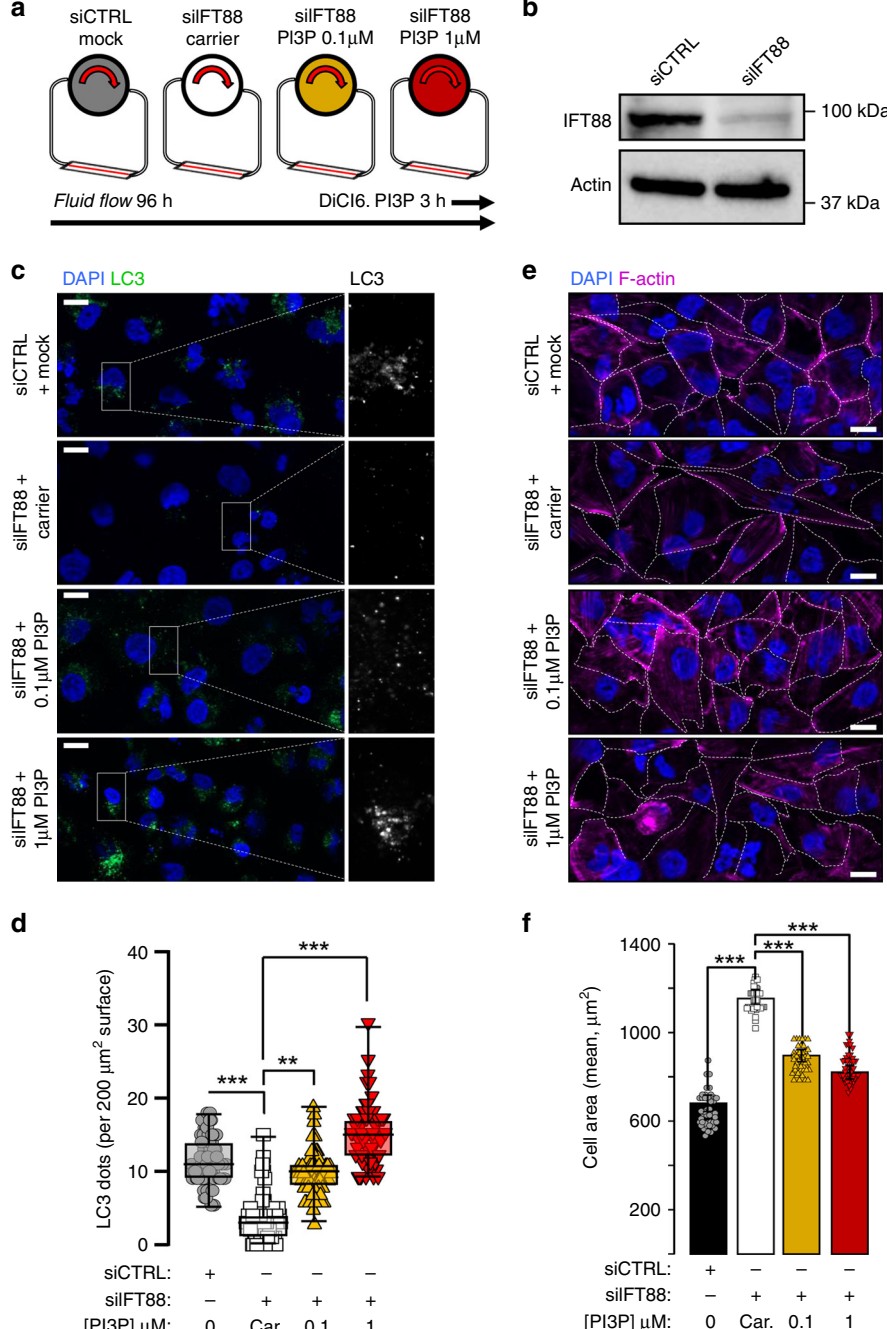

**Fig. 4 Exogenous PI3P delivery restores autophagy and cell size regulation upon shear stress in PC-deprived cells. a** Schematic display of the experimental design; all cells were analyzed after 96 h of fluid flow. We added to IFT88 knockdown (siIFT88) HK2 cells cells two different concentrations of exogenous PI3P (DiCl6.PI3P, in lipid carrier vehicle, at 0.1 and 1 μM) during the last 3 h of fluid-flow treatment and compared them to control cells (siCTRL) and to siIFT88 cells added with carrier alone, as a negative control. **b** Western blot analysis of IFT88 in lysates of HK2 cells (siCTRL) and IFT88 knocked down cells (siIFT88). **c, e** Representative confocal images of HK2 cells (siCTRL), compared to siIFT88, supplemented with exogenous PI3P (0.1 and 1 μM) or with carrier only, upon shear-stress (96 h) conditions, immunostained for LC3 and DAPI or F-actin and DAPI, respectively. Scale bar, 10 μm. **d, f** Quantifications of **c** and **e**. Bar graphs denote average number of LC3 puncta per 200 μm$^2$ area and average cell surface, respectively (mean ± SEM, $N =$ 70, from three independent experiments). $**p < 0.01$, $***p < 0.001$ in two-tailed Student's $t$ test.

conditions) compared to control cells, and the lack of cell volume regulation (Fig. 4e–f, siCTRL and siIFT88 + lipid carrier conditions). Interestingly, external delivery of increasing doses of purified DiCl6.PI3P (0.1 and 1 μM, during the 3 last hours of the fluid-flow treatment) overcame the absence of IFT88, as both readouts were partially recovered (Fig. 4c–f, siIFT88 + PI3P condition), further demonstrating the crucial role of PI3P in the autophagic response induced by shear stress on epithelial cells.

**Overexpression of PI3KC2αrescues absence of PC for autophagy.** Our artificial PI3P delivery experiments showed that presence of PI3P is directly responsible for re-engagement of autophagic program in response to fluid-flow-induced shear stress (Fig. 4). To go further and to connect these data with our observations concerning PI3KC2α mobilization (Fig. 1), requirement for autophagy (Fig. 3) and PI3P-associated autophagy (Fig. 2) during shear-stress sensing, we overexpressed the

human PI3KC2α protein in IFT88 siRNA transfected cells submitted to fluid flow. Interestingly we show that while presence of wild type (WT) PI3KC2α is able to rescue LC3 mobilization as monitored by LC3 lipidation (Supplementary Fig. 11a, b, wild-type condition) and puncta quantification (Supplementary Fig. 11c, d, wild-type condition), the kinase-inactive PI3KC2α (PI3KC2α $K_{inact}$), which lacks the ability to produce PI3P[22,37], was unable to do so (Supplementary Fig. 11a–d, PI3KC2α $K_{inact}$ condition), illustrating the crucial importance of PI3KC2α associated PI3P synthesis in response to shear stress.

## Discussion

The metabolism of phosphoinositides is central to cellular homeostasis regulation in eukaryotic cell. As a prototypical example, PI3P, one of the most abundant lipids of the family, is a central player in endosomal and autophagosomal membrane dynamics[4–6,23], as well as required for PC functions and regulation[19]. PI3P is synthesized by PI3KC3/VPS34, PI3KC2α, PI3KC2β, and PI3KC2γ[38] and despite significant advances in the field, it is still unclear how, where and why these enzymes contribute to different PI3P pools dedicated to specific cellular processes. Here we show that PC-mediated response to mechanical stimulation triggers specific PI3KC2α-dependent PI3P synthesis which participates in autophagic machinery mobilization and in turn in cell volume regulation. More importantly, our data suggest for the first time that two distinct PI3P synthesizing lipid kinases are differentially mobilized, according to the stress type: while PI3KC3/VPS34 is—as expected—crucial for nutritional stress-induced autophagy[11], it is not necessary for the shear-stress-induced autophagic program, which requires instead PI3KC2α activity.

PI3KC2α was reported to be a key regulator of PC functions via the synthesis of a pool of PI3P, which in turn is crucial for Rab11a membrane mobilization and activation[19,28]. Here we demonstrate that PI3KC2α knockdown not only affects ciliogenesis, but also has severe consequences on PC -associated functions and signaling in cells subjected to shear stress, a situation known to activate the autophagic program via the PC machinery[17]. Indeed, we show that Rab11a is stabilized by shear stress in ciliated cells, and that its mobilization is affected upon PI3KC2α knockdown, presumably as a consequence of the PI3KC2α dependent PI3P pool depletion. Our data linking Rab11a activation to shear-stress associated PI3P further confirm recent report supporting an essential role for Rab11a, PI3P-binding protein WIPI2 and associated PI3P-positive recycling endosomal membranes in autophagosome biogenesis[27].

We observe that PI3KC2α stability is strongly increased by shear stress and that artificial delivery of PI3P, or overexpression of PI3KC2α (but not the PI3KC2α kinase-inactive mutant), on PC-deprived polarized cells, compensate for the PC role and is sufficient to reboot the autophagic response associated with shear-stress sensing. Our observation that PI3KC2α is not essential for starvation-induced (VPS34 dependent) autophagy, could explain why the knockdown of Beclin1, the key partner of VPS34 in starvation-associated PI3P synthesis[33], or the knockdown of VPS34 itself, have no deleterious effect on the shear stress-associated response. Thus, our results make sense with previous observation of absence of Beclin1recruitment at the basal body during PC associated autophagy[31]. We therefore propose that class 3 and class 2 PI3kinases are involved in different kinds of autophagic response. Thus, it can be postulated that cellular adaptation to shear stress mobilizes PI3KC2α-dependent PI3P synthesis without depending on VPS34, which in turn is available for endosomal trafficking regulation, as a putative consequence of previously reported increase in shear stress-induced endocytosis[39,40], as well for autophagic response to other stimuli.

Our findings open up the question of canonical versus non-canonical autophagy discrimination in the mechanical stimulation of epithelial ciliated cells. It is indeed tempting to hypothesize that, while PI3P is essential for autophagy-machinery mobilization (notably in the autophagosome biogenesis sequence), the source of PI3P can differ from one stress situation to another, with, at least in our experimental data, exclusive PI3KC2α lipid enzyme mobilization upon shear-stress conditions in relation with the PC. In this context, we would like to emphasize that the low shear stress used in this study (i.e. 1 dyn/$cm^2$) reflects the primary urine flow in normal kidney epithelium, thus suggesting that in vitro studies like the present one contribute to reveal the importance of PI3KC2α activity in a still poorly studied, despite being physiological, situation. In this regard, it is tempting to hypothesize that a specific mobilization of the PI3P-dependent autophagic program may directly participate in the PI3KC2α-mediated prevention of renal cysts formation[20], highlighting the protective role of the autophagy and PC dialog in kidney epithelium homeostasis. PI3KC2α activity is known to be involved in many physiological processes beyond its role in kidney physiology, by notably controlling mast cells degranulation, angiogenesis, thrombosis and systemic glucose homeostasis (reviewed in[41]). Whether shear-stress autophagy is a downstream effector of PI3KC2α in these activities remains to be investigated.

## Methods

**Animals and in vivo samples analysis**. PI3KC2α[+/−] mice were described in Franco et al.[19]. Briefly, a *lacZ* (bacterial β-galactosidase)-*neoR* cassette was inserted in-frame with the ATG start codon of *Pik3c2a* via bacterial recombination. Constructs were electroporated in ES cells and chimeras obtained by standard procedures. Mice were backcrossed for eight generations in the C57Bl/6J. Wild-type littermates from heterozygous crosses were used as controls. Mice were from the breeding facility of CREFRE (US006, Toulouse, France) and maintained under SPF conditions at the animal facility of Rangueil (Anexplo platform, US06, Toulouse, France). All animal experimental procedures were conducted in accordance with institutional guidelines on animal experimentation approved by the local ethical committee of animal care and are conformed to the guidelines from Directive 2010/63/EU of the European Parliament on the protection of animals used for scientific purposes or the NIH guidelines. For morphological analysis and immunohistochemistry, mouse kidneys were fixed in 4% paraformaldehyde and paraffin embedded. For surface quantification, four-micrometer kidney sections were stained with Periodic Acid Shiff (PAS) and at least 50 tubular sections were measured for each mouse using Calopix software (TRIBVN). Briefly, the area of external profile of the tubule and the area of the lumen were measured and epithelial surface was calculated as the difference between the two areas. For autophagy quantification, 4 μm kidney sections were incubated with anti-LC3 (MBL, PM036) antibody, followed by fluorophore-conjugated secondary antibody (Molecular Probes). Kidney sections were incubated with biotinylated-Lotus tetragonolobus lectin (LTL) (AbCys biology) followed by Alexa Fluor® 555 streptavidin (Molecular Probes) to stain proximal kidney tubules. DAPI was used to stain nuclei. Whole kidney sections were scanned using a nanozoomer 2.0 HT (Hamamatsu) with a ×40 oil immersion objective. LC3 *punctae* were quantified using Calopix software (TRIBVN) in all the microscopic cortical fields of the kidney section.

**Cell culture and transfections**. HK2 cells (ATCC) were cultured in Dulbecco's Modified Eagle Medium (DMEM), supplemented with 10% FCS at 37 °C and 5% $CO_2$. For the starvation experiments, cells were incubated with Earle's balanced salt solution (EBSS) for the indicated times. Confluent HK2 cells were transfected with GFP-WIPI2 (a kind.pngt from Tassula Proikas-Cezanne), GFP-PI3KC2α and GFP-PI3KC2α kinase inactive (Franco et al.[19]) cDNAs using Lipofectamine 2000 (Invitrogen, Life Technologies) according to the manufacturer's instructions. SiRNA transfections were performed using Lipofectamine RNAi Max (Invitrogen, Life Technologies) according to the manufacturer's instructions and two siRNA primers were used for each target at a final concentration of 20 nM. All siRNAs were purchased from Qiagen and the references are as follows: Control siRNA (SI1027281); BECN1 (Beclin1) siRNA (SI00055580 and SI00055573); VPS34/PIK3C3 siRNA (SI00040950 and SI00040971); PIK3C2α (SI00040894 and SI00040901); IFT88 (SI04374552, SI04294416, SI03180065 and SI00752374); FIP200 (SI02664578, SI03036194, SI02664571 and SI00108122); IFT20 (SI04132919 and SI03134355).

**Shear-stress induction**. HK2 cells were seeded ($2.25 \times 10^5$ in 150 µl of medium) into a microslide "I0.6 Luer" chamber (channel dimensions: $50 \times 5 \times 0.4$ mm, Ibidi) and cultured for 96 h to allow proper polarization and epithelial differentiation. The microslides were connected to a fluid-flow system, which contains an air-pressure pump and a two-way switch valve that pumps the culture medium uni-directionally between two reservoirs through the flow chamber at a rate corresponding to a shear stress of 1 dyn/cm$^2$. The control cells (static) were set up in the same microslide Luer chambers and maintained in culture as long as the flow-subjected cells.

**Protein extraction, western blotting analysis and antibodies**. Cells in microslides were washed twice with ice-cold PBS and lysed in ice with 150 µl of 1X Laemmli buffer (60 mM Tris-HCL pH = 6.8, 2% SDS, 10% Glycerol, bromophenol blue, supplemented with 100 mM DTT) for 30 min. Samples were boiled for 10 min at 95 °C, separated by SDS/PAGE and then transferred onto Nitrocellulose membranes. Western blot analysis was performed with specific antibodies and the antigen–antibody complexes were visualized by chemiluminescence (Immobilon Western, Merck Millipore). The following antibodies were used in immunoblotting: rabbit-anti LC3 (1:1000, Sigma, Cat#L7543); mouse-anti-actin (1:5000, Millipore, Cat#1501); mouse-anti-Beclin1 (1:2000, BD Biosciences,Cat#612113); rabbit-anti-IFT88 (1:1000, Proteintech, Cat#13967-1-AP); rabbit-anti-PIK3C2α (1:800, Novus, Cat#NBP2-19829); mouse-anti-Wipi2 (2A2) (1:1000, AbD Serotec, Cat#MCA5780GA); rabbit-anti-Rab11 (1:1000, Cell Signaling, Cat#2413S); rabbit-anti-VPS34 (d9A5) (1:500, Cell Signaling, Cat#4263); rabbit-anti-LKB1 (1:1000, Cell signaling, Cat#3050); rabbit-anti-FIP200 (1:1000, Sigma, Cat#SAB4200135); mouse-anti-ATG16L1 (1:1000, MBL, Cat#PM040); rabbit-anti-IFT20 (1:1500, Proteintech, Cat#13615). Secondary HRP conjugate anti-rabbit IgG (GE Healthcare) and HRP conjugate anti-mouse IgG (Bio-Rad).

**Immunofluorescence and microscopy**. Cells were fixed either with 4% paraformaldehyde (PFA) for 20 min or with cold methanol for 5 min at −20 °C for proper PC axoneme proteins detection. Cells were then washed and incubated for 30 min in blocking buffer (10% FCS in PBS) followed by incubation with primary antibodies diluted in blocking buffer supplemented with 0.05% saponin for 1 h at room temperature or overnight at 4 °C. Cells were washed 3 times, and then incubated for 1 h with fluorescent Alexa-Fluor secondary antibodies. After washing, 150 µl of DAPI-Fluoromount were added into the Luer chamber (Southern Biotech). For the labeling of PI3P with FYVE-FYVEGST, cells were fixed and incubated for 1 h with purified FYVE-FYVEGST recombinant protein (20 µg/ml final concentration), washed with PBS, and labeled with a FITC-conjugated anti-GST antibody (Rockland) as fully described[23]. Images were acquired with a Zeiss Apotome.2 fluorescence microscope or Zeiss LSM700 confocal microscope both equipped with 63x oil immersion fluorescence objectives. Number of ciliated cells and length of cilia were quantified using Zen Software (Zeiss) or Imaris Software (Bitplane).The following antibodies were used for immunofluorescence: mouse-anti-LC3B (1:200, MBL, Cat# M152-3); mouse-anti-ARL13B (C-5) (1:200, Santa Cruz, Cat#515784); rabbit-anti-ARL13B (1:200, Proteintech, Cat#515784); rabbit-anti-ATG16L1 (1:100, MBL, Cat#PM040); rabbit-anti-p-AMPK (T172) (1:200, Cell signaling, Cat#2535); rabbit-anti-PIK3C2A (1:500, Novus, Cat#NBP2-19829); mouse-anti-WIPI2 (1:200, Bio-Rad, Cat#MCA5780GA); rabbit-anti-βcatenin (1:500, Cell Signaling, Cat#8480); mouse-anti-γ-Tubulin (1:400, Sigma; Cat#T5326); rabbit-anti-γ-Tubulin (1:500, Sigma; Cat#T5192); mouse-anti-acetylated Tubulin (1:200, Sigma; Cat#T7451); rabbit-anti-LKB1 (1:500, Cell signaling, Cat#3050); mouse-anti- Rab11-GTP (1:100, New East Biosciences; Cat#26919); Phalloidin (1:100, Cat# A34055). Alexa Fluor-conjugated secondary antibodies (donkey anti-mouse IgG and donkey anti-Rabbit IgG, Life Technologies).

**External delivery of purified PI3P**. DiCI6.PI3P ($C_{41}H_{77}Na_3O_{16}P_2$, Echelon) and lipid carrier (shuttle PIP$^{TM}$, Echelon) were reconstituted in $H_2O$ separately following the manufacturer's instructions and then mixed to prepare stock solution (0.5 mM DiC16PI3P). Final mixture was diluted in fresh media for the indicated final concentrations (0.1 and 1 µM) and used on microslide chambers connected to the fluid-flow system and pumps for the three last hours of shear-stress treatment. Carrier only was used as a negative control on siIFT88 transfected HK2 cells.

**Real time quantitative PCR**. RNA was extracted from cells using the NucleoSpin RNA kit (Macherey-Nagel). Reverse transcriptase PCR and qRT-PCR were performed using "Power Sybr green cells to CT" kit (Thermo Fisher Scientific) according to manufacturer's instructions. Actin was used as reference gene and relative quantification was calculated using the ΔΔCT method. Primers sequences are as follows:

PIK3C2α -forward: 5′-TGAATAGTTCATTAGTGCAATTCCTT-3′; PIK3C2α –reverse: 5′-GGCATCTTTGAGAAGCCAAT-3′

WIPI2 -forward: 5′-ACTGGCTACTTTGGGAAGGTTCC-3′; WIPI2 –reverse: 5′-AGATGCAGAGTCTACGAT-3′

Actin-forward: 5′-GGCCAACCGTGAAAAGATGA-3′; Actin-reverse: 5′-ACCAGAGGCATACAGGGACAG-3′

**Statistical analysis**. Data are presented as means ± SD or SEM. Statistical analyses were performed by unpaired, two-tailed Student's $t$ test, using GraphPad Prism7 (*$p < 0.005$, **$p < 0.001$, and ***$p < 0.0001$). Images showing Western blotting or immunofluorescence analysis are representative of three independent experiments unless otherwise stated.

**Reporting summary**. Further information on research design is available in the Nature Research Reporting Summary linked to this article.

## Data availability
The data that support the findings of this study are available from the authors on reasonable request.

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

## Acknowledgements

The authors are grateful to our colleagues at Institut Necker Enfants Malades (France) and Institute of Metabolic and Cardiovascular Diseases (Toulouse) for critical reading of the manuscript. The authors warmly thank Dr Veronique Pons (INSERM UMR 1048) for help and advices on PI3KC2α constructs. This study was supported in part by ANR (agence nationale de la recherche), INSERM (institut national de la santé et de la recherche médicale), CNRS (centre national de la recherche scientifique), Université de Paris, Fondation de France and Fédération Française de Cardiologie.

## Author contributions

A.B. and A.C.N. performed most of the biological experiments and analyses, D.R. and N. D. performed and analyzed in vivo experiments, E.H. contributed to wild-type and kinase-inactive PIK3C2α overexpression experiments and corrected the manuscript, S.G. contributed to in vivo experiments and general experimental design and corrected the manuscript, M.L. co-supervised the project and corrected the manuscript, P.C. co-supervised the project and wrote the paper and E.M. contributes to lipid delivery and imaging experiments, analyzed and quantified the cell biological assays and fluorescence data, supervised the project and wrote the paper.

## Competing interests

The authors declare no competing interests.
