## [Peer Review File · Nature Communications]

Reviewers' comments:

Reviewer #1 (Remarks to the Author):

In the at hand manuscript Nascimbeni et al. investigate the PI3KC2a dependent generation of PIP3 and its role in the modulation of autophagy in response to shear stress in ciliated epithelial cells. The authors show that PI3KC2a expression increases upon shear stress and that this enzyme generates PI3P at the vicinity of the primary cilium (PC). Knockdown of PI3KC2a abolishes autophagy and cell volume adaption to shear stress. PI3P generation near the PC is PI3KC2a dependent and purified PI3P can rescue the autophagic response to shear stress after the siIFT88-mediated PC loss.

This is an interesting follow up story on a recent study by the authors in NCB in 2016 suggesting a mechanism to their previous finding linking cilia and autophagy. While this is an interesting study, there are some issues that should be addressed by the authors:

1. Fig. 1: While the authors describe "increased stability" of PI3KC2a, the data (Fig 1a/b) allow only the conclusion, the expression is increased, both of RNA and protein. This should be corrected in the result section. Ciliary localization of PI3KC2a should be quantified. Is PI3KC2a present in all cilia?
2. In Fig 1e the exact localization of PIP3 remains unclear. 3D confocal images should be presented and analyzed. Images should be quantified from a larger number of cilia.
3. Figure 2a/b/c: Why did the authors choose this large area of 150 μm^2 , including the nucleus (where PI3P is also localized)? This makes it a very rough measurement. Which size of Z is applied? Wouldn't an IF staining with pericentrin/ γ -Tubulin and the measurement of the PC basal body area give a more precise measurement, like is done in Fig 1F? Additionally, how was the measurement done after loss of PC in siPI3KC2a cells? Suppl. Fig 2c illustrates a dramatic effect on ciliogenesis, where the remaining 10% ciliated cells might be the ones escaping siRNA transfection. By the way, is this a loss of ciliogenesis or a loss of Arl13b positive cilia?
4. Fig 4a: how efficient does the partial knockdown of IFT88 really abrogate ciliogenesis? Does this remove cilia completely or result in shortened cilia? To allow the conclusion, that the observed effect is cilia dependent and not related to any other IFT88 function, ciliogenesis should be inhibited by an additional independent approach. In Fig 4c/e cilia should be co-stained.
5. In Suppl Fig 2a and Suppl Fig 4a the same western blot is presented - with Suppl Fig 4 somehow showing the complete picture. Different pictures should be chosen or both figures should be combined.

Minor points:

- 1) Abstract "We show that PI3KC2a depletion in epithelial cells subjected to shear stress remarkably abolish ciliary function as well as the autophagic response and related cell size regulation in vitro and in vivo." Abolishment of all ciliary functions is not investigated. Did authors mean cilium formation? Or a specific function? Please rephrase.
- 2) P.10 "specific PI3P pool" Are the authors sure this is a specific pool, and not relocation of protein from other cellular compartment(s)?
- 3) Discussion: Could the authors comment on the (clinical) application and impact of their findings?
- 4) Fig 2d) bar graphs/legend inconsistent. Please correct. Y-axis seems to be normalized ration rather than A.U.

Reviewer #2 (Remarks to the Author):

In this manuscript Nascimbeni and co-workers report that the PI3K class II α (PI3KC2a) is responsible for generation of a pool of PI3P at the base of the primary cilium upon fluid flow-

induced shear stress and that depletion of PI3KC2 α prevents induction of autophagy and primary cilia formation under conditions of shear stress. In contrast, depletion of the PI3KC3/VPS34 kinase, which is critical for starvation-induced autophagy, did not affect shear stress-dependent autophagy.

The data presented in this manuscript are well done and nicely presented, but unfortunately not of sufficient novelty to warrant publication in Nature Communication. It has previously been shown (although not under shear stress) that PI3KC2 α is enriched at the base of the primary cilium, where it regulates production of a specific pool of PtdIns3P, and that its depletion cause defects in primary cilium elongation, which can be rescued by reconstitution of PI3P levels (Franco et al, Dev Cell 2014). Moreover, it is known that fluid flow-induced shear stress leads to induction of autophagy and cell size reduction (Orhon et al, NCB 2016). Thus, the authors should aim at obtaining further insight into the mechanisms involved in the differential regulation of PI3KC2 α and VPS34 localization and/or activation under shear stress conditions. It might also be useful to address the following comments:

Figure 2d: how is the level of WIPI2 regulated upon shear stress – at transcriptional or protein stability level? Why is total level of WIPI2 and RAB11 decreased upon siPI3KC2 α ?

Is the autophagy machinery required for shear stress-induced autophagy the same as for starvation-induced autophagy? Is WIPI2b recruited to the PI3P-positive structures at the base of the primary cilium?

Fig S4a-b; seems like LC3-II is reduced by siVPS34 in the blot, but not in the corresponding quantifications

Reviewer #3 (Remarks to the Author):

In this paper, the authors suggest that PI3KC2 α , a lipid enzyme required for ciliogenesis and the proper function of the PC, promotes the synthesis of a local pool of PI3P upon fluid flow-induced shear stress. Because understanding the mechanism to regulate PI3KC2 α by shear stress is very important, the article is really interesting. It is well established that cilia regulates autophagy and that autophagy is involved in the cell size regulation. Yet, the role PI3KC2 α is still unknown in particular in relation to shear stress. Overall the study is well done and of general interest, however, there are a few issues to be solved.

Questions:

In Fig1c,e, PI3Kc2 α localizes to cilia depend on the shear stress, but Why does PI3P not localize to cilia?

In Fig1c, PI3KC2 α partially localize to the cilia, but the localization is not specific. Where does PI3KC2 α localize in the organelle? Furthermore, quantification of the cilia-positive ratio per total PI3KC2 α would be helpful to make these data convincing.

Can the authors speculate about the mechanisms by which the shear stress promotes the PI3KC2 α relocalisation in response to shear?

In cilia-deficient condition, is the expression of PI3P and PI3KC2 α affected? and does shear stress

induce the PI3KC2a expression?

Not sure I understand the labels and quantifications in Fig1d,f, and e. What happens to gamma tubulin when shear is present? The staining looks much fainter than in regular conditions. How is this affecting the quantification shown in 1f?

The authors have to show the data about the percentage of cilia-positive HK2 cells before- and after-treatment of shear stress.

Can the total concentration of PI3P calculate before- and after-treatment of shear stress?

Fig 2, if the authors would like to mention the cilia-dependent PI3P accumulation, it is essential to compare the PC region and the other cytoplasmic area. For example, the authors are able to measure the total area per cell and the percentage of cilia-region.

In Fig3, it looks like 24h shear can induce the LC3 formation, implying that the PI3P is already accumulated in the cell, however the authors have treated shear stress for 96 h in Fig 1. Why does this condition differ from Fig1?

How does PI3KC2a mutant data compare the Vps34 mutant mouse data?

The related-question in Fig 1, PI3KC2a transfection can rescue induce the autophagy in siIFT88 condition in Fig4c, e?

Reviewer #1 (Remarks to the Author):

In the at hand manuscript Nascimbeni et al. investigate the PI3KC2a dependent generation of PIP3 and its role in the modulation of autophagy in response to shear stress in ciliated epithelial cells. The authors show that PI3KC2a expression increases upon shear stress and that this enzyme generates PI3P at the vicinity of the primary cilium (PC). Knockdown of PI3KC2a abolishes autophagy and cell volume adaption to shear stress. PI3P generation near the PC is PI3KC2a dependent and purified PI3P can rescue the autophagic response to shear stress after the silFT88-mediated PC loss.

This is an interesting follow up story on a recent study by the authors in NCB in 2016 suggesting a mechanism to their previous finding linking cilia and autophagy.

We are thankful to this reviewer for her/his positive comment concerning our findings on the importance of PI3KC2a in the primary cilium/autophagy machineries dialog during shear stress adaptation.

While this is an interesting study, there are some issues that should be addressed by the authors:

1. Fig. 1: While the authors describe “increased stability” of PI3KC2a, the data (Fig 1a/b) allow only the conclusion, the expression is increased, both of RNA and protein. This should be corrected in the result section.

We thank the reviewer for this precision: the results section of the manuscript has been modified accordingly.

Ciliary localization of PI3KC2a should be quantified. Is PI3KC2a present in all cilia?

This reviewer raised an important point: from the experiments we made, it is rather difficult to claim that PI3KC2a is present all through the axoneme. By using different cell fixation methods, we were able to show that PI3KC2a is present at the base of the cilium (notably by PFA fixation) and at both the base of the cilium and at discrete locations associated with the axoneme (by cold methanol fixation). As suggested by this reviewer, we quantified the presence of PI3K at the PC, from 3D confocal acquisitions (Figure 1d and 1e).

2. In Fig 1e the exact localization of PIP3 remains unclear. 3D confocal images should be presented and analyzed. Images should be quantified from a larger number of cilia.

As requested by this reviewer, we now show 3D confocal acquisitions and quantifications of the PI3P abundance at the base of the PC (Figure 1g and 1h).

3. Figure 2a/b/c: Why did the authors choose this large area of 150 μm^2 , including the nucleus (where PI3P is also localized)? This makes it a very rough measurement. Which size of Z is applied? Wouldn't an IF staining with pericentrin/ γ -Tubulin and the measurement of the PC basal body area give a more precise measurement, like is done in Fig 1F? Additionally, how was the measurement done after loss of PC in siPI3KC2 α cells?

We apologize if the method chosen to quantify PI3P structures at the vicinity of the PC was not clear enough. We decided to quantify the presence of PI3P-positive dots in the area where PI3P appears at the base of the PC (such as seen in Figure 1g for example) in the shear situation. As ciliogenesis is itself perturbed in cells depleted of PI3KC2a (Supplementary Figure 2), we have chosen other PC markers (such as ARL13B or acetylated tubulin) to locate the "150 μm^2 area" and quantify PI3P in siPI3KC2a conditions. This is better illustrated now in Figure 2a, and quantified in Figure 2c.

Suppl. Fig 2c illustrates a dramatic effect on ciliogenesis, where the remaining 10% ciliated cells might be the ones escaping siRNA transfection. By the way, is this a loss of ciliogenesis or a loss of Arl13b positive cilia?

As proposed by this reviewer, it is likely that the remaining ciliated cells observed in our experiments are the ones escaping siRNA transfection. Importantly, we now show that the knock down of PI3KC2a definitively affects ciliogenesis itself, and not only ARL13B expression, since we now observed similar results using acetylated tubulin, another axoneme marker, as illustrated in Supplementary Figure 2d.

4. Fig 4a: how efficient does the partial knockdown of IFT88 really abrogate ciliogenesis? Does this remove cilia completely or result in shortened cilia? To allow the conclusion, that the observed effect is cilia dependent and not related to any other IFT88 function, ciliogenesis should be inhibited by an additional independent approach. In Fig 4c/e cilia should be co-stained.

We chose to work with IFT88 downregulation since this protein is known to be essential for ciliogenesis (Pazour et al, J Cell Biol, 2000). As suggested by this reviewer, we are now fully documenting the effects of IFT88 siRNA on HK2 cells: supplementary Figure 10 shows that IFT88 knockdown strongly diminishes ciliogenesis (sup Figure 10a and 10b). Similar results were obtained using knockdown of IFT20, another key protein involved in ciliogenesis (Jurczyk et al, J Cell Biol, 2004), as compared in supplementary Figure 10c to 10g, where we notably show that LC3 lipidation in response to shear stress is impaired both in silFT88 and silFT20 conditions. Interestingly, we also show that the upregulation of PI3KC2a that we observe as a consequence of shear stress (Figure 1a) is no longer present in cells knocked-down for IFT88 (sup Figure 10h and 10i). Finally we now show that absence of IFT88 also leads to a strong decrease in the PI3P mobilization at the base of PC in response to shear stress (sup Figure 10j and 10k). Altogether, these data sustain the choice we made to use IFT88 knockdown to analyze the effect of exogenous delivery of purified PI3P on cells lacking a proper ciliogenesis and submitted to shear stress, as shown in Figure 4.

5. In Suppl Fig 2a and Suppl Fig 4a the same western blot is presented - with Suppl Fig 4 somehow showing the complete picture. Different pictures should be chosen or both figures should be combined.

We apologize for this mistake in the editing of western blot pictures and we thank the reviewer for letting us know. We corrected the figures accordingly (supplementary Figure 2a and (new) supplementary Figure 7a).

Minor points:

1) Abstract “We show that PI3KC2 α depletion in epithelial cells subjected to shear stress remarkably abolish ciliary function as well as the autophagic response and related cell size regulation in vitro and in vivo.” Abolishment of all ciliary functions is not investigated. Did authors mean cilium formation? Or a specific function? Please rephrase.

We indeed intended to say that ciliogenesis was affected, as well as signaling cascades which are associated with PC in response to shear stress, such as P-AMPK and LKB1. We rephrased the abstract accordingly.

2) P.10 “specific PI3P pool“ Are the authors sure this is a specific pool, and not relocalization of protein from other cellular compartment(s)?

Our observations led us to think that specific PI3P synthesis occurs as part of an autophagy-associated shear stress response, which is more in favor of a “specific pool” of PI3P associated with the process. However, we cannot exclude that pre-existing PI3P-positive structures could be also re-localized by the signaling cascade occurring during autophagic response to shear stress. Accordingly, we rephrased this conclusion.

3) Discussion: Could the authors comment on the (clinical) application and impact of their findings?

We included a short paragraph concerning PI3KC2a physiological and pathophysiological implications processes.

4) Fig 2d) bar graphs/legend inconsistent. Please correct. Y-axis seems to be normalized ration rather than A.U.

We corrected the figure accordingly.

Reviewer #2 (Remarks to the Author):

In this manuscript Nascimbeni and co-workers report that the PI3K class II α (PI3KC2 α) is responsible for generation of a pool of PI3P at the base of the primary cilium upon fluid flow-induced shear stress and that depletion of PI3KC2 α prevents induction of autophagy and primary cilia formation under conditions of shear stress. In contrast, depletion of the PI3KC3/VPS34 kinase, which is critical for starvation-induced autophagy, did not affect shear stress-dependent autophagy.

The data presented in this manuscript are well done and nicely presented, but unfortunately not of sufficient novelty to warrant publication in Nature Communication. It has previously been shown (although not under shear stress) that PI3KC2 α is enriched at the base of the primary cilium, where it regulates production of a specific pool of PtdIns3P, and that its depletion cause defects in primary cilium elongation, which can be rescued by reconstitution of PI3P levels (Franco et al, Dev Cell 2014). Moreover, it is known that fluid flow-induced shear stress leads to induction of autophagy and cell size reduction (Orhon et al, NCB 2016). Thus, the authors should aim at obtaining further insight into the mechanisms involved in the differential regulation of PI3KC2 α and VPS34 localization and/or activation under shear stress conditions.

We thank this reviewer for her/his positive comments about our manuscript and for the associated suggestions to increase the novelty of our observations. As detailed below, we now present a substantial amount of data which strengthen our conclusions about the key role of PI3KC2 α , compared to VPS34, in primary cilium-associated autophagic response to shear stress. We notably show that while PI3KC2 α is indeed required for proper ciliogenesis (Supplementary figure 2) absence of VPS34 has no effect on primary cilium dynamics, whatever the flux situation (Supplementary figure 3). Moreover, we now show that early regulator of autophagy FIP200, a key partner of ULK1 known to actively regulate the VPS34/Beclin1 complex during starvation-induced autophagy, is not mobilized by shear stress and not required for shear stress-associated autophagic response (Supplementary figure 9). It is interesting to note that while many ATG proteins have been found at the PC vicinity in response to shear stress, neither ULK1 partner FIP200 nor Beclin1 were found associated with PC (Pampliega et al, Nature, 2013). These data make sense with our previous results concerning the lack of Beclin1 requirement in the specific autophagic response induced by shear stress (Supplementary figure 8) and strengthen the differential regulation, depending on the type of stress, of PI3P synthesis source (VPS34 vs PI3KC2 α) that we report in our present manuscript. In addition, we now show that the autophagic crucial protein ATG16L1, which was recently demonstrated to be an essential partner of both Rab11

and WIPI2, was directly affected in absence of PI3KC2a-driven PI3P synthesis in response to shear stress (Supplementary figure 6).

Finally, we now show that PI3KC2a is directly required for autophagic response to shear stress by overexpressing the kinase in cells in which ciliogenesis was impaired (see supplementary Figures 10 and 11) by knocking down the IFT88 ciliary protein. Interestingly, we show that while the wild type PI3KC2a expression was sufficient to rescue autophagic response in absence of primary cilium, the kinase-inactive mutant of PI3KC2a had no effect, thus demonstrating that the lipid kinase activity of PI3KC2a was directly required to enable autophagic process induced by shear stress through local PI3P synthesis (Supplementary Figure 11).

It might also be useful to address the following comments:

Figure 2d: how is the level of WIPI2 regulated upon shear stress – at transcriptional or protein stability level? Why is total level of WIPI2 and RAB11 decreased upon siPI3KC2a?

We thank the reviewer for this remark. Our new data indeed suggest that WIPI2 levels are regulated at transcriptional level. We now show that the levels of WIPI2 mRNA are not affected by absence of PI3KC2a in static conditions, but they increase after 4 days of shear stress only in cells expressing PI3KC2a, and not in cells knocked-down for PI3KC2a (Figure 2e). Moreover, we now report that ATG16L1, the key-partner of ATG5-12 conjugate required for proper LC3 lipidation during autophagosome biogenesis, was directly affected by absence of PI3KC2a (i.e. by absence of PI3KC2a associated PI3P synthesis) only in response to shear stress; interestingly, total amount of ATG16L1 is strongly increased by shear stress stimulation (Supplementary Figure 6a and 6b), and targeted at the base of the primary cilium under the same conditions (Supplementary Figure 6c), a phenomenon which is no longer observed in cells knock-down for the PI3KC2a enzyme (Supplementary Figure 6a, 6b and 6c). ATG16L1 was demonstrated recently to interact with both Rab11 and WIPI2 (the latter being responsible for autophagy-associated PI3P binding) during autophagy initiation: our results thus suggest that absence of shear stress-induced and PI3KC2a-dependent PI3P leads to destabilization of WIPI2/ATG16L1/Rab11 dialog, a situation which is likely to block the autophagic process at early steps, leading to absence of LC3 lipidation and mobilization, as we initially observed in our report (Figure 3 and Supplementary Figure 7).

Is the autophagy machinery required for shear stress-induced autophagy the same as for starvation-

induced autophagy? Is WIPI2b recruited to the PI3P-positive structures at the base of the primary cilium?

This reviewer raised a very important point. First, and based on this reviewer's suggestion, we now report that the pool of PI3P specifically mobilized at the base of the primary cilium under shear stress (see new Figure 2a-c) is indeed associated with WIPI2, as illustrated by WIPI2/PI3P colocalization at the base of primary cilium observed in shear stress situation (Figure 2f).

Moreover, we now studied the requirement of "classical / starvation induced autophagy" machinery in the shear stress-associated situation: making sense with our data demonstrating that both VPS34 (Figure 3) and Beclin1 (Supplementary Figure 8) are not required for shear stress/primary cilium autophagy, we now show that FIP200, a crucial member of the ULK1 autophagy initiation complex required during starvation, was neither mobilized by shear stress (Supplementary Figure 9a and 9b) nor required for autophagic response to shear stress (Supplementary Figure 9c and 9d). This last result clearly demonstrates that during shear stress response, both the Vps34/Beclin1 complex and its upstream regulatory complex ULK1 are not necessary for autophagy, since PI3KC2a is able to drive specific PI3P synthesis instead.

Fig S4a-b; seems like LC3-II is reduced by siVPS34 in the blot, but not in the corresponding quantifications

The levels of LC3-II indeed display a slight decrease tendency (Supplementary Figure 7a) in siVPS34 cells but, from our data, the differences were not significant (Supplementary Figure 7b), and, more importantly, not comparable with the LC3 lipidation decrease observed in siPI3KC2a cells under shear stress.

Reviewer #3 (Remarks to the Author):

In this paper, the authors suggest that PI3KC2 α , a lipid enzyme required for ciliogenesis and the proper function of the PC, promotes the synthesis of a local pool of PI3P upon fluid flow-induced shear stress. Because understanding the mechanism to regulate PI3KC2a by shear stress is very important, the article is really interesting. It is well established that cilia regulates autophagy and that autophagy is involved in the cell size regulation. Yet, the role PI3KC2a is still unknown in particular in relation to shear stress. Overall the study is well done and of general interest, however, there are a few issues to be solved.

We wish to thank this reviewer for these encouraging and positive comments concerning the interest of our article and about the overall quality of the study.

Questions

In Fig1c,e, PI3Kc2a localizes to cilia depend on the shear stress, but why does PI3P not localize to cilia?

This is indeed an important question; in fact we are able to detect only a fraction of PI3KC2a in the cilium axoneme, and only in cells fixed with cold methanol (see new 3D data on Figure 1d and 1e). Due to technical reasons, it is possible to detect endogenous PI3P by fluorescence only in cells fixed with PFA, but not with methanol. Since methanol fixation only allows proper axonemal components detection, we cannot rule out the possibility that a small amount of PI3P is as well present in the axoneme. Nevertheless, we now show a new set of data, including quantifications, that clearly show accumulation of PI3P-positive structures at the base of the PC (Figure 1g, Figure 2a and 2c), which are as well positive for the PI3P-binding autophagic regulator WIPI2 (Figure 2f).

In Fig1c, PI3KC2a partially localize to the cilia, but the localization is not specific. Where does PI3KC2a localize in the organelle? Furthermore, quantification of the cilia-positive ratio per total PI3KC2a would be helpful to make these data convincing.

We thank this reviewer for the suggestion. We now show, and quantified, PI3KC2a distribution at the PC using 3D reconstruction (Figure 1d and 1e) from Z-stacks confocal acquisitions.

Can the authors speculate about the mechanisms by which the shear stress promotes the PI3KC2a relocalisation in response to shear?

It is tempting to speculate that PI3KC2a is recruited to Rab11 positive vesicles / positive membranes at the base of the PC during shear sensing by the PC. We indeed show that Rab11, which is known to have a privileged partnership with PI3KC2a at the vicinity of PC (Franco et al, Dev Cell, 2014), is clearly associated with the response to shear stress driven autophagy (supplementary figure 5). Moreover, we now show that ATG16L1, a key autophagic regulator, is as well mobilized by shear stress at the base of the PC under shear stress sensing (see Sup Figure 6). This ATG16L1 mobilization was abolished in cells lacking PI3KC2a (i.e. in cells lacking the PI3KC2a-associated PI3P pool synthesized in response to shear stress) (Sup Figure 6). Because ATG16L1 is a direct partner of both Rab11 (Puri et al, Dev Cell, 2018) and WIPI2 (Vicinanza et al, Oncotarget, 2019; Dooley et al. Mol Cell, 2014), which we found colocalized on PI3P-positive structures at the base of PC under shear situations (Figure 2f), we can imagine that PI3KC2a targeting and stabilization at PC depends on Rab11 as well as WIPI2 and ATG16L1, which can confer the autophagic machinery specificity of the system.

In cilia-deficient condition, is the expression of PI3P and PI3KC2a affected? and does shear stress induce the PI3KC2a expression?

This was a very useful suggestion: we now show that in IFT88-depleted cells (in which ciliogenesis is abolished (Supplementary Figure 10a and 10b)) the shear stress is no longer leading to PI3KC2a mobilization (Supplementary Figure 10h and 10i) and does not stimulate local synthesis of PI3P at the immediate vicinity of the remaining PC structure (Supplementary Figure 10j and 10k). This strengthens the inter-relationship between PI3KC2a regulation and PC-associated machinery and turnover in response to shear stress.

Not sure I understand the labels and quantifications in Fig1d,f, and e. What happens to gamma tubulin when shear is present? The staining looks much fainter than in regular conditions. How is this affecting the quantification shown in 1f?

We apologize for the choice of the illustrations in our initial manuscript. Accordingly, we now show new illustrations of the 3D confocal acquisitions that were used for the PI3P-positive structures we quantified (Figure 1g).

The authors have to show the data about the percentage of cilia-positive HK2 cells before- and after-treatment of shear stress.

We previously published that the percentage of ciliated HK2 cells is not significantly affected in shear stress compared to control, while the length of cilia was increased (Zemirli et al, 2019 (see Figure 2C, 2D and 2E, “siCTRL situations”)). We are now citing our previous results more adequately in the present manuscript.

Can the total concentration of PI3P calculate before- and after-treatment of shear stress?

These data are shown in the Supplementary Figure 7c and 7d and we can clearly observe an increase in the total number of PI3P-positive vesicles in cells submitted to shear stress.

Fig 2, if the authors would like to mention the cilia-dependent PI3P accumulation, it is essential to compare the PC region and the other cytoplasmic area. For example, the authors are able to measure the total area per cell and the percentage of cilia-region.

We apologize if we were not clear enough about the quantification methods we used. We indeed, as this reviewer suggests, did show both the “PC region”-associated presence of PI3P vesicles (notably shown in Figure 1g and 1h, Figure 2a, b and c and supplementary Figure 10j and k) but also those from the total cytoplasmic area, as shown in supplementary Figure 7c and 7d.

In Fig3, it looks like 24h shear can induce the LC3 formation, implying that the PI3P is already accumulated in the cell, however the authors have treated shear stress for 96 h in Fig 1. Why does this condition differ from Fig1?

We apologize if we were not clear enough. Indeed, we classically used 96h of shear stress to detect maximal autophagic response, even though we are able to detect LC3 lipidation after shorter periods of

time (Supplementary Figure 1a and Zemirli et al, 2019), but in the data presented in Fig3, we aimed at comparing directly the autophagic response induced by starvation or shear stress. 96h of starvation was a too long period of time and such a treatment would have killed the cells. We thus decided to limit both autophagic stimulations we wished to compare to 24h.

How does PI3KC2a mutant data compare the Vps34 mutant mouse data?

We would like to thank the reviewer for giving us the opportunity to discuss this point. In fact, the specific kidney invalidation of PI3KC2a has not been reported so far, in contrast to the VPS34 conditional knock out (Grieco et al 2018 Sci Rep)). Nevertheless, we provide below the comparison of the proximal tubule phenotype between the full PI3KC2a +/- mouse model and the conditional invalidation of Vps34 in the proximal tubule. As far as the proximal tubule is considered, only results from a full PI3KC2a +/- have been reported (Franco et al 2015 J Am Soc Nephrol). A defect in the primary cilium elongation and a defect of polycystine 2 localization at the primary cilium have been observed. These animals are prone to develop kidney cysts in response to ischemia/reperfusion. In parallel, a conditional invalidation of VPS34 has been reported in kidney proximal tubules (Grieco et al 2018 Sci Rep). These animals developed a Fanconi-like syndrome, a kidney failure and died. Thus the proximal tubule phenotype of VPS34 mice is distinct from that observed in the proximal tubule of PI3KC2a +/- mice. With regard to autophagy, no data were reported in the Pik3c2a +/- model except those reported in our manuscript. Here, it is interesting to note that cells with cysts have an enlargement of cell size (Boehlke et al 2010 Nat Cell Biol). In light of our present data we can speculate that this phenotype is related to a defect in autophagy downstream of PI3KC2a. In the VPS34 model no impairment of autophagosome formation was observed but an impairment of lysosome positioning that finally alters the autophagic flux (Grieco et al 2018 Sci Rep). These results suggest that VPS34 is not required for autophagosome formation in the proximal tubule. At this point it should be emphasized that the deletion of Vps34 in mice has a tissue-specific effect on autophagy as reported in the literature (e.g. autophagosomes are not formed in heart and liver but still observed in motor neurons).

The related-question in Fig 1, PI3KC2a transfection can rescue induce the autophagy in siFT88 condition in Fig4c, e?

This was a key suggestion for our manuscript revision: we now indeed show that in cells with altered ciliogenesis (*i.e.* IFT88 knock down cells (see supplementary Figure 10)), the exogenous expression of PI3KC2a was sufficient to rescue autophagic response; more importantly, we show that this was not the case anymore when we expressed the kinase inactive mutant of PI3KC2a (see Supplementary Figure 11), thus demonstrating that the kinase activity of PI3KC2a was directly required to enable autophagic process through local PI3P synthesis in response to shear stress.

REVIEWERS' COMMENTS:

Reviewer #1 (Remarks to the Author):

In this revised manuscript the authors addressed all of my concerns. Therefore, I strongly support publication of this study.

Reviewer #2 (Remarks to the Author):

The authors have very nicely responded to my initial comments and concerns and the manuscript is now acceptable for publication.

Reviewer #3 (Remarks to the Author):

The authors addressed all my points. This is a great and important study.